# Efficacy and moderators of efficacy of trauma-focused cognitive behavioural therapies in children and adolescents: protocol for an individual participant data meta-analysis from randomised trials

Anke de Haan [1,2,3] Caitlin Hitchcock [1] Richard Meiser-Stedman [4] Markus A Landolt [2,3] Isla Kuhn [5] Melissa J Black [1,6] Kristel Klaus [1] Shivam D Patel [1] David J Fisher,[7] Tim Dalgleish [1,6]

For numbered affiliations see end of article.

**Correspondence to**
Dr Anke de Haan;
Anke.deHaan@mrc-cbu.cam.ac.uk

## ABSTRACT

**Introduction** Trauma-focused cognitive behavioural therapies are the first-line treatment for posttraumatic stress disorder (PTSD) in children and adolescents. Nevertheless, open questions remain with respect to efficacy: why does this first-line treatment not work for everyone? For whom does it work best? Individual clinical trials often do not provide sufficient statistical power to examine and substantiate moderating factors. To overcome the issue of limited power, an individual participant data meta-analysis of randomised trials evaluating forms of trauma-focused cognitive behavioural therapy in children and adolescents aged 6–18 years will be conducted.

**Methods and analysis** We will update the National Institute for Health and Care Excellence guideline literature search from 2018 with an electronic search in the databases PsycINFO, MEDLINE, Embase, Cochrane Central Register of Controlled Trials and CINAHL with the terms (trauma* OR stress*) AND (cognitive therap* OR psychotherap*) AND (trial* OR review*). Electronic searches will be supplemented by a comprehensive grey literature search in archives and trial registries. Only randomised trials that used any manualised psychological treatment—that is a trauma-focused cognitive behavioural therapy for children and adolescents—will be included. The primary outcome variable will be child-reported posttraumatic stress symptoms (PTSS) post-treatment. Proxy-reports (teacher, parent and caregiver) will be analysed separately. Secondary outcomes will include follow-up assessments of PTSS, PTSD diagnosis and symptoms of comorbid disorders such as depression, anxiety-related and externalising problems. Random-effects models applying restricted maximum likelihood estimation will be used for all analyses. We will use the Revised Cochrane Risk of Bias tool to measure risk of bias.

**Ethics and dissemination** Contributing study authors need to have permission to share anonymised data. Contributing studies will be required to remove patient identifiers before providing their data. Results will be published in a peer-reviewed journal and presented at international conferences.

**PROSPERO registration number** CRD42019151954.

### Strengths and limitations of this study

► This is the first individual participant data meta-analysis (IPD-MA) of trauma-focused cognitive behavioural therapies in children and adolescents.
► In contrast to existing individual studies, an IPD-MA will provide the statistical power to examine moderating factors of trauma-focused cognitive behavioural therapies in children and adolescents.
► Only randomised controlled trials will be included to allow us to evaluate the efficacy of trauma-focused cognitive behavioural therapies over and above the non-specific effects of comparator conditions.
► A variety of measures of the primary and secondary outcomes will have been used in the individual studies bringing commensurate methodological and statistical complexity.
► Study findings will enhance the future provision and development of trauma-focused cognitive behavioural therapies in children and adolescents.

## INTRODUCTION

Within the last two decades, research in children and adolescents has tremendously increased our knowledge about trauma-related disorders such as posttraumatic stress disorder (PTSD), the long-lasting impact of potentially traumatic events (PTEs) and the efficacy of trauma-focused therapies in younger populations. Trauma-focused cognitive behavioural therapies are the first-line treatment for PTSD in children, adolescents and adults.[1] They are a category of

psychological interventions including trauma-focused cognitive behavioural therapy,[2] cognitive therapy for PTSD,[3] prolonged exposure therapy for adolescents[4] and the child-friendly version of narrative exposure therapy (KidNET[5]) (see the recent guideline from the National Institute for Health and Care Excellence (NICE)[6]).

Classic meta-analyses synthesising aggregated data from randomised controlled trials (RCTs) have shown that trauma-focused cognitive behavioural therapies are effective in reducing psychological distress including PTSD in children and adolescents.[7–11] However, open questions remain with respect to clinical outcome: why does this first-line treatment not work for everyone? For whom does it work best? Factors that might impact the efficacy of trauma-focused cognitive behavioural therapies in children and adolescents form two broad categories: treatment-related and child-related factors. Treatment-related factors may include the length of therapy, involvement of parents in the intervention, and the balance of behavioural and cognitive intervention components. Child-related factors may include the type of trauma, the severity of symptoms, comorbid diagnoses, gender, age and other trauma-related or demographic variables. Current stand-alone RCTs invariably lack the power to explore the contribution of these factors to clinical outcomes, and have produced a mixed pattern of findings (eg, see references[12–17]). Further, classical meta-analysis, due to its reliance on summary data, is typically unable to comprehensively evaluate such moderating factors.

To overcome these problems of limited power we propose an individual participant data meta-analysis (IPD-MA) of randomised trials. By addressing the critical question about what works for whom, we hope to enhance the future provision and development of trauma-focused cognitive behavioural therapies in children and adolescents. In a first step, our aim is to determine the efficacy of trauma-focused cognitive behavioural therapies for children and adolescents, relative to control and active comparison conditions. A second step then addresses our central aim to explore moderators of treatment effects, both treatment-related factors and child-related factors. Both of these aims are theory-driven and of high clinical relevance for successfully treating children and adolescents who have been exposed to trauma. The following hypotheses will be examined:

▶ Hypothesis 1: trauma-focused cognitive behavioural therapies will produce a greater reduction in posttraumatic stress symptoms (PTSS) in children and adolescents compared with either (1) no intervention (no treatment, waitlist), (2) treatment-as-usual (TAU), (3) individual non-trauma focused psychosocial interventions or (4) other individual trauma-focused psychosocial interventions.

▶ Hypothesis 2a: efficacy of trauma-focused cognitive behavioural therapies will be significantly predicted by predefined treatment-related factors available at trial baseline. Due to the mixed findings from previous studies, non-directional hypotheses will be tested. Post-treatment PTSS will be significantly predicted by:

– Predefined intended length of treatment (number of sessions).
– Predefined intended involvement of caregivers.

▶ Hypothesis 2b: child-related factors will serve as prognostic predictors for the efficacy of trauma-focused cognitive behavioural therapies. Due to the mixed findings from previous studies, non-directional hypotheses will be tested. Post-treatment PTSS will be significantly predicted by:

– Age of the participants at the start of treatment.
– Gender.
– Trauma-type of index-event.
– Trauma-history.
– Severity of PTSS pretreatment.

## METHODS AND ANALYSIS
### Study registration and management
This IPD-MA will be conducted in accordance with Preferred Reporting Items for a Systematic Review and Meta-Analysis of Individual Participant Data (the PRISMA-IPD statement[18]). Regular email updates will be sent to inform the collaborating network of study progress. End-to-end encrypted electronic data-sharing clouds and email will be used to exchange pseudo-anonymised data and paperwork between researchers.

### Patient and public involvement
Patients were not involved in this study. However, secondary data analysis ensures maximum return from patient involvement in research. The outcome of this IPD-MA will be published in an international peer-reviewed journal. The findings will further be presented at international conferences.

### Ethics and dissemination
We will cite the ethics code for each contributing study in the published paper. Contributing studies will be required to remove patient identifiers before providing their data. This includes names, addresses, and date of birth which will be converted to age-at-index-trauma-event and age-at-time-of-assessment. Contributing studies will need to have permission to share anonymised data.

### Criteria for included studies
#### Types of studies
Only randomised studies will be included in this IPD-MA. Articles must be written in English. Unpublished data will be actively sought; hence, non-peer-reviewed studies will also be included. We will perform sensitivity analyses to evaluate the impact of published versus unpublished studies on our results.

#### Participants
Studies must have recruited children and adolescents aged 6–18 years exposed to a single-event trauma (eg, road traffic accident) or multi-event trauma (eg, domestic violence) sufficient to meet the DSM-IV or DSM-5 definitions of a

de Haan A, *et al. BMJ Open* 2021;**11**:e047212. doi:10.1136/bmjopen-2020-047212

qualifying traumatic event. We will request studies with a broader age range; however, only participants within our defined age range will be included in the IPD-MA. Sensitivity analyses will be conducted if the adult version of a treatment were administered to an adolescent sample. A standardised outcome measure comprising either a diagnostic interview indexing symptom severity or a self-report measure of PTSS must have been administered before and after treatment. Furthermore, a clinically relevant degree of severity of PTSS at trial baseline must have been present as assessed either by scoring above a validated cut-off on a PTSS rating scale or by meeting criteria for PTSD. We will also request studies that include both children and adolescents with and without clinically relevant severity of PTSS as defined earlier. However, again, only those participants with clinically relevant severity of PTSS will be included in the analyses.

### Treatments

In line with the NICE guideline,[6] we will include studies that used any manualised psychological treatment that we deem to be a trauma-focused cognitive behavioural therapy for children and adolescents. This includes cognitive therapy, cognitive processing therapy, compassion focused therapy, exposure therapy/prolonged exposure, virtual reality exposure therapy, imagery rehearsal therapy and KidNET. We furthermore adopt NICE guideline description of trauma-focused cognitive behavioural therapy as laid out in the associated paper (Mavranezouli *et al*,[10] p19); namely, 'a broad class of psychological interventions that predominantly use trauma-focused cognitive, behavioural or cognitive behavioural techniques and exposure approaches to treatment. Although some interventions place their main emphasis on exposure (eg, imaginal reliving, producing a written narrative or in vivo exposure) and others on cognitive techniques (eg, restructuring of trauma-related appraisals), most use a combination'. Independent raters will evaluate author descriptions of their treatment with respect to this definition to determine inclusion within the IPD-MA. In contrast to the NICE guideline,[6] we will not include mindfulness-based cognitive therapy as a trauma-focused cognitive behavioural therapy.

In addition, treatment may be delivered in-person or online, but must comprise an individual-, rather than group-format, and a multi-session treatment protocol. A minimum of at least one post-treatment/ follow-up assessment must have been reported.

### Comparison conditions

Trauma-focused cognitive behavioural therapies will be compared: (1) against no intervention (no treatment, waitlist); (2) against TAU; (3) against individual non-trauma focused psychosocial interventions or (4) against other individual trauma-focused psychosocial interventions. Again, comparison condition type will be determined by two independent raters, based on the author descriptions.

### Primary outcomes

The primary outcome variable will be child-reported PTSS using a standardised self-report post-treatment (see the Strategy for data synthesis section for further information). Proxy-reports (teacher, parent and caregiver) will be analysed separately. The primary endpoint of post-treatment will be indexed as the assessment completed immediately after completion of trauma-focused cognitive behavioural therapy, less than 1 month after the final treatment session.

### Secondary outcomes

Secondary outcomes will include: (1) follow-up assessments of PTSS; (2) PTSD diagnoses and (3) symptoms of comorbid disorders such as depression, anxiety-related and externalising problems, reported via self-reports and proxy-reports. Follow-up lengths to be included comprise assessments between 1 month and 2 years following the completion of therapy. During analysis, studies including a follow-up assessment between 1 and 3 months post-treatment will be grouped to form a short-term follow-up, and any later assessment points will be grouped per 6-month period (ie, 6 months, 12 months). This will result in analysis of follow-up outcomes in the short-term (1–3 months), and at 6, 12, 18 and 24 months post-treatment.

### Search methods for identification of studies and obtaining datasets

Figure 1 depicts our multi-layered search method in order to obtain all potential studies, published and unpublished (in line with the PRISMA-IPD statement[18]).

### Electronic searches

Publications identified by the latest NICE guideline for PTSD published in 2018 will be included (NICE guideline search was completed on 29 January 2018). To update the results of the NICE search, an electronic search using the same databases will be restricted to publications between the 1 January 2018 and 12 November 2019. We will replicate the NICE guideline search by using the same search terms related to trauma-focused cognitive behavioural therapies. We will exclude specific search terms that are not related to a psychosocial intervention and thereby unlikely to include a cognitive-behavioural therapy approach. This includes physiotherapy and biological interventions (eg, transcranial magnetic stimulation). We will also exclude specific terms defining any intervention that is not cognitive behaviour therapy-based, for example, hypnosis or dance therapy. Finally, we will remove terms referring to occupational/return to work support.

Searches in the databases PsycINFO via EBSCOhost, MEDLINE via Ovid, Embase via Ovid, Cochrane Central Register of Controlled Trials (CENTRAL) via Cochrane Library, CINAHL via EBSCOhost will use the following search plan: (trauma* OR stress*) AND (cognitive therap* OR psychotherap*) AND (trial* OR review*)

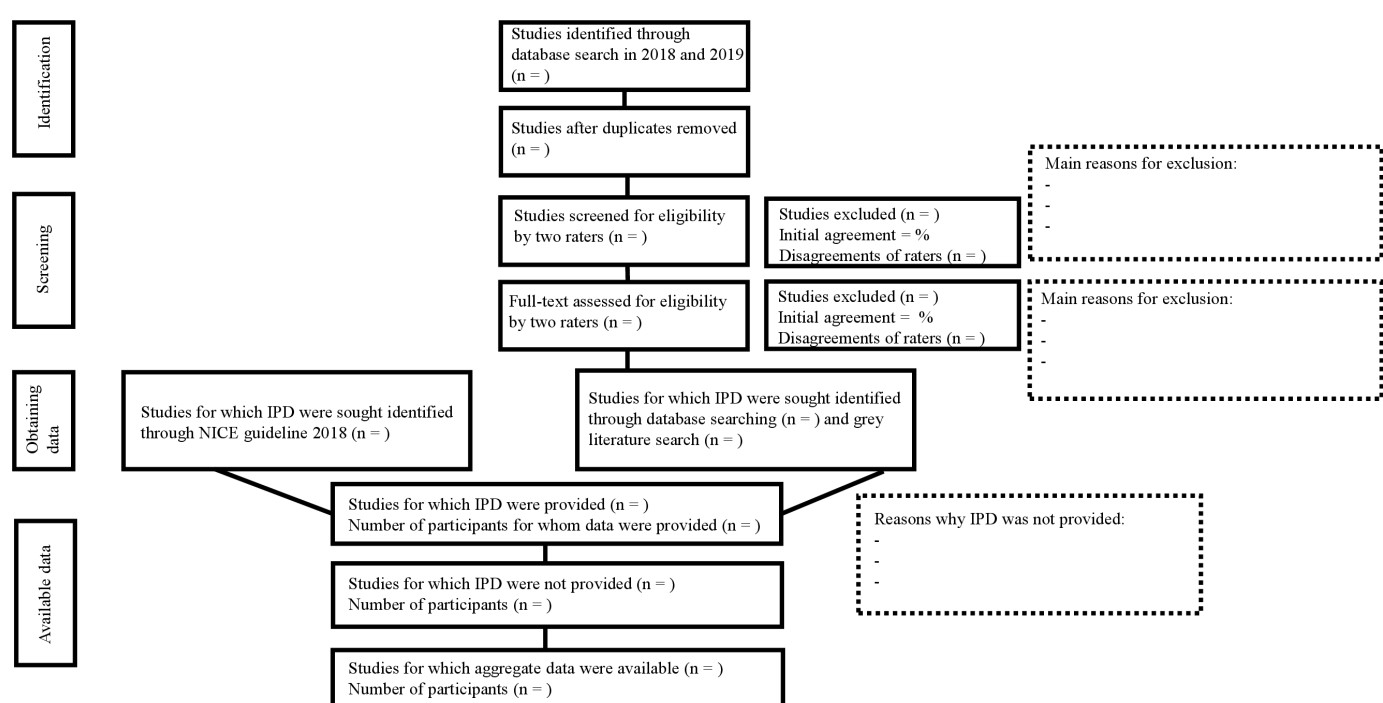

**Figure 1** Flow-chart showing data acquisition. IPD, individual participant data; NICE, National Institute for Health and Care Excellence.

(see online supplemental material 1 for the full search strategy).

No language restrictions will be applied to the search, but only studies published in English will be included. Search results will be deduplicated in Endnote (by IK), then imported into Rayyan (a web-based tool for managing systematic reviews[19]). This will allow for blind screening by raters. Two raters will independently review title and abstract of the records identified in the electronic search. The raters will review the records according to the following exclusion criteria applying the same exclusion order: (1) duplicate; (2) language other than English; (3) review or meta-analysis; (4) no randomised trial; (5) no applicable age range extractable; (6) no manualised trauma-focused cognitive behavioural therapy; (7) group-format; (8) single-session treatment; (9) no assessment post-treatment; (10) no standardised outcome measure to assess PTSS; (11) no clinically relevant PTSS extractable; (12) comparison condition outside protocol. Records deemed ineligible from title and abstract by both raters will be set aside. Records appearing eligible (ie, not meeting any exclusion criterion) or where eligibility can not be determined due to insufficient information in the abstract will proceed to the full-text stage. Again, at the full-text stage, two raters will examine the remaining records independently. Any disagreements will be resolved via discussion with RM-S, MAL and TD.

### Grey literature

Clinical trial registries and archives will be searched up to the 12 November 2019 using the following search string: (child* OR adolesc* OR youth OR young*) AND (PTSD OR posttraumatic stress disorder). We will use the trial registries ClinicalTrials.gov and ISRCTN to identify any relevant unpublished trials, including those that are currently ongoing. Moreover, the archives PsyArXiv and bioRxiv will be searched to identify any relevant preprints up to 6 months prior (12 May 2019) of the electronic search (12 November 2019). Finally, we will check reference lists of included studies and relevant meta-analyses identified by the electronic search to make sure that all available trials will have been detected by the NICE guideline, our electronic search, and grey literature search.

### Non-literature based searching

Key authors will be contacted via email to request any unpublished datasets, and Twitter will be used to raise awareness of the IPD-MA.

### Data collection

Corresponding authors of eligible studies will be emailed to request data. A reminder email will be sent after two weeks. If an author does not respond after two emails, another author of the study will be contacted as well (either first, second or last author). A second attempt to contact both authors together will follow. A maximum of three authors per article will be contacted. We will consider study data unavailable if no study authors respond to multiple contact attempts, or if authors indicate that they no longer have access to the data or do not wish to make their data available. A single person for each included study will be designated to whom all queries about the data collection processes and transformation of individual variables will be addressed. When cleaning and preparing a specific data set, communication with

de Haan A, et al. BMJ Open 2021;11:e047212. doi:10.1136/bmjopen-2020-047212

**Table 1** Individual participant data to be extracted from included studies

| Treatment-related factors | Child-related factors | Outcomes |
|---|---|---|
| *Descriptives* | *Demographics* | |
| Trial identifier | Anonymised participant identifier | |
| Country of completion | Gender | |
| Information about risk of bias | Age | |
| Type of trauma-focused cognitive behavioural therapy | Ethnicity | |
| Type of comparison group/s | Trauma type of index-event | |
| Number of sessions | Trauma history | |
| Length of treatment in weeks | | |
| Involvement of caregivers | | |
| Any potential covariates (eg, mode of administration, profession of therapists) | Any potential covariates (eg, pretreatment levels of dysfunctional posttraumatic cognitions, IQ, social support, treatment expectancy, therapeutic alliance) | Any related outcome variables (eg, post-treatment level of dysfunctional posttraumatic cognitions and changes in coping behaviours) |
| | *Psychological symptoms pre-treatment* | *Psychological symptoms post-treatment and follow-up* |
| | Pre-treatment self-reported and proxy-reported PTSS | Self-reported and proxy-reported PTSS post-treatment and follow-up |
| | Pre-treatment self-reported and proxy-reported depression symptoms | Self-reported and proxy-reported depression symptoms post-treatment and follow-up |
| | Pre-treatment self-reported and proxy-reported anxiety-related symptoms | Self-reported and proxy-reported anxiety-related symptoms post-treatment and follow-up |
| | Pre-treatment self-reported and proxy-reported externalising problems | Self-reported and proxy-reported externalising problems post-treatment and follow-up |
| | *Diagnoses pre-treatment* | *Diagnoses post-treatment and follow-up* |
| | Pre-treatment diagnostic status of PTSD | Diagnostic status of PTSD post-treatment and follow-up |
| | Pre-treatment diagnostic status of comorbid disorders | Diagnostic status of comorbid disorders post-treatment and follow-up |
| *Reason for missing data* | *Reason for missing data* | *Reason for missing data* |

PTSD, posttraumatic stress disorder; PTSS, posttraumatic stress symptoms.

the original investigators will take place by email or telephone.

## Data extraction, quality checks and storage

The primary variables to be requested from study investigators are listed in table 1. We aim to collect data on all of these variables from all studies, regardless of whether such data were previously published. For all outcomes, unimputed and untransformed data will be requested. Data will be cleaned and stored separately for each study. Spot checks will be completed to ensure data quality. The pattern of treatment allocation for each included study will be checked to ensure that randomisation and allocation sequence appear appropriate, in accordance with guidelines recommended by Tierney *et al.*[20] For final

checks before analysis and the statistical analyses, the datasets will be combined into a single dataset. Data will be stored in password-protected files on an encrypted University of Cambridge server.

## Risk of bias

Two raters will independently evaluate the risk of bias for the included studies by using the revised Cochrane Risk of Bias tool (RoB 2[21]) to access study quality and risk of bias due to the randomisation process, deviations from intended interventions, missing outcome data, measurement of the outcome and selection of the reported result. Each study will be rated as of high risk, some concerns, or low risk.

## Strategy for data synthesis

Data will be analysed across a series of stages, in order to be guided by data availability and the degree of potential for harmonisation. Unimputed and untransformed data will be requested.

First, for published studies, key variables will be re-analysed within each study (eg, participant numbers per treatment condition, mean PTSS scores pretreatment and post-treatment, numbers gender, mean age), as to identify any potential inconsistencies in the supplied data.

Second, data will be harmonised as far as possible: (1) the definitions and scales of outcomes (eg, standardising PTSS total scores across different measures); (2) the timings of measurements (eg, pretreatment defined as the assessment directly before start of treatment; post-treatment defined as the assessment immediately after completing treatment, less than 1 month after the final treatment session); (3) the definitions, scales and/ or subgroups used for covariates (eg, the specific index trauma event will be grouped into accidental trauma, natural disaster, war trauma, or interpersonal trauma).

Third, depending on the amount of missing data and whether missing at random assumptions are met, multiple imputation will be carried out.

Based on the stepwise approach described earlier, decisions may be made for example, to put aside certain desired analyses or adjustment factors if it is felt that data are too limited and may bias results. Once the final constitution of the model has been agreed based on the above, we will proceed to the meta-analysis pooling itself. Note that modelling will be done for each outcome (as specified in table 1) separately. All analyses will be completed using random-effects models employing restricted maximum likelihood estimation. However, if there is considerable heterogeneity in the quality of studies (indexed by RoB 2), a sensitivity analysis will be completed comparing random-effects and fixed-effects models.

Depending on the IPD data sets we receive, we will collect aggregate data (AD) from studies where IPD could not be obtained and combine it with the IPD to tackle inclusion bias.[22] In this case, sensitivity analyses will be performed comparing an IPD-only meta-analysis with a meta-analysis that combines IPD and AD.

One-stage approaches will be applied using the R software.[23] We will investigate the overall summary of treatment effect and we are further interested in the heterogeneity in treatment effect across-studies and within-studies. If the one-stage model fails to converge, a two-stage model will be calculated.

### Hypothesis 1

A one-stage linear mixed effects (LME) model with random intercept and outcome baseline adjustment with different residual variance per study will be applied to analyse the effect of trauma-focused cognitive behavioural therapies on the continuous outcome of PTSS post-treatment. A mixed effects logistic regression model will be used to analyse the treatment effect on the binary outcome PTSD diagnosis post-treatment.

Sensitivity analysis will be completed contrasting a (single) random intercept (as primary) with a separate fixed intercept for each study (as sensitivity). As described earlier, if the one-stage model fails to converge, a two-stage model will be completed.

### Hypothesis 2a

We will use meta-regression to explore whether the overall effect of trauma-focused cognitive behavioural therapies varies in relation to treatment-related factors such as predefined intended length of treatment and predefined intended involvement of caregivers.

### Hypothesis 2b

To investigate whether the overall effect of trauma-focused cognitive behavioural therapies varies by child-related factors such as age, gender, trauma type of index-event, trauma-history and symptom severity pretreatment, subject-level interactions will be investigated. Interaction term between treatment status and subject-level covariates will be specified.[22]

Random-effects distributions for the interaction effects will be specified. Effects on the continuous primary outcome of PTSS post-treatment will be analysed using a one-stage LME model with random intercept to account for correlation between the interaction estimate and other parameter estimates. Patient-level covariates will be centred to separate within-trial and across-trial effects. A mixed effects logistic regression model for the binary secondary outcome of PTSD diagnosis post-treatment will be applied. Sensitivity analysis will be completed contrasting a (single) random intercept (as primary) with a separate fixed intercept for each study (as sensitivity). As described earlier, if the one-stage model fails to converge, a two-stage model will be completed.

### Additional analyses with subsamples

Depending on the data provided, we will investigate further treatment-related factors such as predefined mode of administration, profession of therapists and treatment expectancy pretreatment. Moreover, additional child-related factors such as comorbidity pretreatment, pretreatment levels of dysfunctional posttraumatic cognitions, IQ and pretreatment parental mental health will be addressed.

### Future analyses

In future investigations of the obtained data, we plan to conduct mediation analyses within the IPD-MA context to evaluate mechanisms of action of trauma-focused cognitive behavioural therapies. The most promising candidates seem to be changes in targeted cognitive and behavioural processes; for example, improvements in dysfunctional posttraumatic cognitions of the child regarding being permanently and disturbingly changed, or feeling vulnerable,[24–26] as well as changes in safety-seeking behaviours.[25]

Moreover, we are aiming to investigate non-responding, deterioration and possible predictors for dropping out

de Haan A, *et al. BMJ Open* 2021;**11**:e047212. doi:10.1136/bmjopen-2020-047212

from trauma-focused cognitive behavioural therapies across studies.

**Author affiliations**

[1]Medical Research Council Cognition and Brain Sciences Unit, University of Cambridge, Cambridge, UK

[2]Department of Psychology - Division of Child and Adolescent Health Psychology, University of Zurich, Zurich, Switzerland

[3]Department of Psychosomatics and Psychiatry, University Children's Hospital Zurich, Zurich, Switzerland

[4]Department of Clinical Psychology and Psychological Therapies, Norwich Medical School, University of East Anglia, Norwich, UK

[5]Medical Library, University of Cambridge, Cambridge, UK

[6]Cambridgeshire and Peterborough NHS Foundation Trust (CPFT), Cambridge, UK

[7]MRC Clinical Trials Unit at UCL, Institute of Clinical Trials and Methodology, University College London, London, UK

**Contributors** AdH, CH, RM-S, MAL and TD designed the project. IK, MJB, KK, SDP and DJF contributed to study methods. All authors critically reviewed and approved the final manuscript.

**Funding** AdH is funded by the Swiss National Science Foundation (Grant Reference: P2ZHP1_187612). CH and SDP are partly supported by the Economic and Social Research Council (Grant Reference: ES/R010781/1). MJB is partly supported by the National Institute for Health Research Cambridge Biomedical Research Centre. KK is funded by the UK Medical Research Council (Grant Reference: MC_PC_17213). DJF is funded by the UK Medical Research Council (Grant Reference: MC_UU_12023/24). TD is funded by the UK Medical Research Council (Grant Reference: SUAG/043 G101400) and partly supported by the National Institute for Health Research Cambridge Biomedical Research Centre.

**Competing interests** None declared.

**Patient consent for publication** Not required.

**Provenance and peer review** Not commissioned; externally peer reviewed.

**ORCID iDs**

Anke de Haan http://orcid.org/0000-0002-4676-348X

Caitlin Hitchcock http://orcid.org/0000-0002-2435-0713

Richard Meiser-Stedman http://orcid.org/0000-0002-0262-623X

Markus A Landolt http://orcid.org/0000-0003-0760-5558

Isla Kuhn http://orcid.org/0000-0002-2879-4020

Melissa J Black http://orcid.org/0000-0003-1450-1140

Kristel Klaus http://orcid.org/0000-0002-3554-3879

Shivam D Patel http://orcid.org/0000-0003-3633-0637

Tim Dalgleish http://orcid.org/0000-0002-7304-2231

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
