## [Reviewer comments · BMJ Open]

ARTICLE DETAILS

TITLE (PROVISIONAL)	Efficacy and moderators of efficacy of trauma-focused cognitive behavioural therapies in children and adolescents – protocol for an individual participant data meta-analysis from randomized trials
AUTHORS	de Haan, Anke; Hitchcock, Caitlin; Meiser-Stedman, Richard; Landolt, Markus; Kuhn, Isla; Black, Melissa; Klaus, Kristel; Patel, Shivam; Fisher, David; Dalgleish, Tim

VERSION 1 – REVIEW

REVIEWER	Bernd Puschner Dept. of Psychiatry II, Ulm University, Germany
REVIEW RETURNED	25-Sep-2020

GENERAL COMMENTS	This is a protocol for an individual patient data (IPD) meta-analysis of trauma-focused cognitive behavioral treatment (CBT) for children and adolescents. It is worthwhile to focus on moderators of effect of such treatments. However, the planned study has considerable problems. Conceptual: While arguing that individual RCTs lack power for moderator analyses, primary research question is still the overall efficacy of CBT for young people. This has been already examined by a large number of meta-analyses. The rationale for doing this again, even with IPD, remains unclear. Methodological: The authors fail to provide a clear definition of moderators, ignoring a considerable bulk of theoretical work since the seminal paper of Baron and Kenny (1986). A common definition of a moderator has been provided by Kraemer et al. (2002, p. 879): “To show that M is a moderator of treatment, M must be a baseline or prerandomization characteristic ... that can be shown to have an interactive effect with treatment on the outcome.” While the authors rightly state that “Factors that might impact the efficacy of trauma-focused cognitive behavioural therapies in children and adolescents form two broad categories: treatment-related and child-related factors.”, by definition, treatment-related factors cannot be moderators, but would rather be mediators (which surprisingly the authors explicitly do not set out to analyze: “In future investigations of the obtained data, we plan to conduct mediation analyses...”). Further comments: • To wonder “Why does this first-line treatment not work for everyone?” seems odd, as there is hardly any such treatment for any condition.• The authors should explain better how they plan to make use of the expected results. It is highly unlikely that merely presenting
--

	results at conferences and publishing a paper will contribute to “enhance the future provision and development of trauma-focused cognitive behavioural therapies in children and adolescents.” References Baron, R. M., & Kenny, D. A. (1986). The moderator–mediator variable distinction in social psychological research: Conceptual, strategic, and statistical considerations. Journal of Personality and Social Psychology, 51(6), 1173. Kraemer, H. C., Wilson, G. T., Fairburn, C. G., & Agras, W. S. (2002). Mediators and moderators of treatment effects in randomized clinical trials. Arch Gen Psychiatry, 59(10), 877–883. https://doi.org/10.1001/archpsyc.59.10.877
REVIEWER	André SCherag Institute of Medical Statistics, Computer and Data Sciences Jena University Hospital – Friedrich Schiller University Jena
REVIEW RETURNED	28-Sep-2020
GENERAL COMMENTS	The authors describe a protocol for an individual participant data meta-analysis from randomized trials. I have three main methodological concerns:  1) The primary outcome is not very well defined as no particular instrument is provided (it is good that a time window is provided); maybe the authors can describe how they will deal with this discrepancy. 2) What are the plans to deal with the multiple comparisons situation in case of multiple control groups? 3) What are the plans to deal with imputed data in the analysis?

VERSION 1 – AUTHOR RESPONSE

Reviewer: 1

Reviewer Name: Bernd Puschner

This is a protocol for an individual patient data (IPD) meta-analysis of trauma-focused cognitive behavioral treatment (CBT) for children and adolescents. It is worthwhile to focus on moderators of effect of such treatments. However, the planned study has considerable problems.

1. Conceptual: While arguing that individual RCTs lack power for moderator analyses, primary research question is still the overall efficacy of CBT for young people. This has been already examined by a large number of meta-analyses. The rational for doing this again, even with IPD, remains unclear.

Our response: *The Reviewer is correct that there are already a few, well conducted meta-analyses of trauma-focused CBT for young people. This is why we described in the title and introduction of our submission that the twin aims of efficacy and moderation are the focus. We think that the Introduction makes it transparent throughout that of these two aims the moderation questions are the most important and novel as they are where IPD-MA has its clearest advantage over traditional meta-analyses. We set out our primary and secondary aims (not research questions) for the IPD-MA in the order that the analyses need to be conducted, not in an order representing relative importance. However, it may be*

that primary and secondary are not the optimal terms and, if so, we are sorry for the confusion caused. We have therefore changed the manuscript as follows (see page 6, paragraph 1):

“In a first step, our aim is to determine the efficacy of trauma-focused cognitive behavioural therapies for children and adolescents, relative to control and active comparison conditions. A second step then addresses our central aim to explore moderators of treatment effects, both treatment-related factors and child-related factors. Both of these aims are theory-driven and of high clinical relevance for successfully treating children and adolescents who have been exposed to trauma.”

Regarding our reason for looking at efficacy at all – something the Reviewer questions. This is standard practice for an IPD-MA, especially where any previous meta-analyses have been on aggregate data. The reason is that an IPD-MA can return a different answer to the aggregate meta-analyses for a number of reasons, e.g., in an IPD-MA we can use multiple imputation to deal with missing data and therefore do intent-to-treat analyses (as opposed to the often-reported analyses on observed data) for all of the trials – something that is not easily possible in an aggregate meta-analysis. These refinements therefore improve the quality of the efficacy estimate.

2. Methodological: The authors fail to provide a clear definition of moderators, ignoring a considerable bulk of theoretical work since the seminal paper of Baron and Kenny (1986). A common definition of a moderator has been provided by Kraemer et al. (2002, p. 879): “To show that M is a moderator of treatment, M must be a baseline or prerandomization characteristic ... that can be shown to have an interactive effect with treatment on the outcome.” While the authors rightly state that “Factors that might impact the efficacy of trauma-focused cognitive behavioural therapies in children and adolescents form two broad categories: treatment-related and child-related factors.”, by definition, treatment-related factors cannot be moderators, but would rather be mediators (which surprisingly the authors explicitly do not set out to analyze: “In future investigations of the obtained data, we plan to conduct mediation analyses...”).

Our response: *We are familiar with this important literature and do agree with this definition. We did not define a moderator as we presumed this was common knowledge in this field. It is of course easy to do so if the reviewer feels that this would benefit the reader. We await editorial guidance.*

*We also agree that some treatment-related factors are not moderators. However, there are many such factors that can be examined for moderation, e.g., a group treatment versus individual, online versus face-to-face therapy, a highly-trained therapist versus an entry-level therapist etc. For our moderation analyses, we specifically note the treatment-related factors we are examining as initial moderators: the intended length of the treatment (e.g. is this a 3-session or a 10-session treatment manual) and whether the treatment is child-only or child-plus-caregiver. These are both variables **available at baseline** as part of the treatment information that is pre-specified in the manualised*

protocol. They are therefore **suitable for moderation analyses**. Further, these factors do not change during treatment, so could not act as mediators as the reviewer suggests. Other treatment variables (e.g. how many actual sessions attended, etc.) are of course not moderators as the reviewer notes. To avoid any confusion, we specified our treatment-related variables as follows (see page 6, paragraph 3):

“Hypothesis 2a: Efficacy of trauma-focused cognitive behavioural therapies will be significantly predicted by pre-defined treatment-related factors available at trial baseline. Due to the mixed findings from previous studies, non-directional hypotheses will be tested. Post-treatment PTSS will be significantly predicted by:

- *Pre-defined intended length of treatment (number of sessions).*
- *Pre-defined intended involvement of caregivers.”*

The same applies for paragraph 2 on page 17:

“We will use meta-regression to explore whether the overall effect of trauma-focused cognitive behavioural therapies varies in relation to treatment-related factors such as pre-defined intended length of treatment and pre-defined intended involvement of caregivers.”

Moreover, we revised the paragraph “Additional analyses with sub-samples” on page 18 to make sure that no confusion is caused: “Depending on the data provided, we will investigate further treatment-related factors such as pre-defined mode of administration, profession of therapists, and treatment expectancy pre-treatment. Moreover, additional child-related factors such as comorbidity pre-treatment, pre-treatment levels of dysfunctional posttraumatic cognitions, IQ, and pre-treatment parental mental health will be addressed.

We agree that in this protocol we do not make an “explicit” commitment to analyse mediators. The reason is that it is unclear until we conduct our stated analyses whether the requirements for mediation (as set out in the very same papers the reviewer cites) will be met. We therefore state that these are for future consideration.

Further comments:

3. To wonder “Why does this first-line treatment not work for everyone?” seems odd, as there is hardly any such treatment for any condition.

Our response: *Sadly, this is true. However, trauma-focused CBT has been advocated as the gold standard for treating children and adolescents across developmental stages, trauma experiences, and cultural background for years. Individual studies often focus only on improvements, but might neglect or might be underpowered to consider factors that predict unsuccessful treatment trajectories.*

Considering the theoretical meta-level of a meta-analysis and the increased power, we feel that it is important to state and consider the very relevant clinical question of treatment non-response.

4. The authors should explain better how they plan to make use of the expected results. It is highly unlikely that merely presenting results at conferences and publishing a paper will contribute to “enhance the future provision and development of trauma-focused cognitive behavioural therapies in children and adolescents.”

Our response: *We agree that translating research findings to clinical practice needs to be prioritized in clinical research. We have been fortunate that our clinical research presented at conferences and published in clinical journals has enhanced future clinical provision (e.g. in NICE guidelines). Of course, lots of other aspects of good implementation science require attention and we will try to deliver our results to the trauma professional community as good as possible. If the reviewer wishes, we are happy to add further possible ways of translating our research findings into practice.*

Reviewer: 2

Reviewer Name: André Scherag

1. The primary outcome is not very well defined as no particular instrument is provided (it is good that a time window is provided); maybe the authors can describe how they will deal with this discrepancy.

Our response: *As the reviewer notes, we carefully define the time-point and the fact that the primary outcome will be a self-report of symptom severity. Alas, it is not possible to be more precise than this in a protocol. The reason is that for an IPD-MA that collates across many trials, the exact outcome instrument will inevitably vary from trial to trial (it would be lovely if everyone used the same instruments but they do not). We do in fact state this in the paper (see page 4):*

"A variety of measures for the primary and secondary outcomes will have been used in the individual trials with commensurate methodological and statistical complexities."

We describe in the section on synthesis how we will harmonise across different instruments from different trials. We now directly refer to the relevant paragraph to make it easier to follow (see page 9, paragraph 4; see paragraph “Strategy for data synthesis” on page 15):

“The primary outcome variable will be child-reported PTSS using a standardised self-report post-treatment (see paragraph “Strategy for data synthesis” for further information).”

Moreover, we aligned our wording throughout the manuscript to self-report and proxy-report. We hope that these generic expressions make clear that we are not able to provide a particular instrument.

2. What are the plans to deal with the multiple comparisons situation in case of multiple control groups?

Our response: This is a very interesting question. It is not relevant for our main questions as stated in the study protocol, where we compare any kind of trauma-focused CBT with any kind of comparison group. However, we do agree that this is a relevant question for a small number of the likely trials if we separate our comparison groups into active comparison groups vs. waitlist/ no treatment. There is not an agreed state-of-the-art procedure in IPD-MA to correct for multiple comparisons. However, the Cochrane handbook makes some suggestions and a recent paper for aggregate meta-analysis provides methodology we believe we can apply to address this issue (<https://doi.org/10.1002/jrsm.1259>). We are happy to insert further details on this, if the reviewer feels that this would benefit the reader. We await editorial guidance.

3. What are the plans to deal with imputed data in the analysis?

Our response: As stated in the manuscript on page 15, paragraph 5, we will use multiple imputation, dependent as usual on the amount of data missing and assumptions of missing-at-randomness. Is there any additional information we should add?

Minor additional revisions

While working on this revision, we realized that the risk of bias paragraph (page 15, paragraph 1) did not use the correct terminology for the Revised Cochrane risk-of-bias tool for randomized trials (RoB 2). It now reads as follows: "Each study will be rated as of high risk, some concerns, or low risk."

Moreover, in line with our treatment paragraph on page 8, we now mention Prolonged Exposure Therapy instead of Cognitive Processing Therapy in the first paragraph of the introduction to better display the variety of trauma-focused cognitive behavioural therapies (see page 5): "They are a category of psychological interventions including Trauma-focused Cognitive Behavioral Therapy (Tf-CBT[2]), Cognitive Therapy for PTSD (CT for PTSD[3]), Prolonged Exposure Therapy for Adolescents (PE-A[4]), and the child-friendly version of Narrative Exposure Therapy (KidNET[5]) (see the recent guideline from the National Institute for Health and Care Excellence [NICE][6])."

4. Foa EB, Chrestman K, Gilboa-Schechtman E. *Prolonged exposure manual for children and adolescents suffering from PTSD*. New York, NY, US: Oxford University Press 2008.